# Optineurin deficiency impairs autophagy to cause interferon beta overproduction and increased survival of mice following viral infection

**Masaya Fukushi**[1]*, **Ryosuke Ohsawa**[2], **Yasushi Okinaka**[3], **Daisuke Oikawa**[4], **Tohru Kiyono**[5], **Masaya Moriwaki**[2], **Takashi Irie**[1], **Kosuke Oda**[1], **Yasuhiro Kamei**[6], **Fuminori Tokunaga**[4], **Yusuke Sotomaru**[7], **Hirofumi Maruyama**[8], **Hideshi Kawakami**[2], **Takemasa Sakaguchi**[1]

1 Department of Virology, Graduate School of Biomedical and Health Sciences, Hiroshima University, Hiroshima, Japan, 2 Department of Epidemiology, Research Institute of Radiation Biology and Medicine, Hiroshima University, Hiroshima, Japan, 3 Graduate School of Integrated Sciences for Life, Hiroshima University, Higashi-Hiroshima City, Hiroshima, Japan, 4 Department of Medical Biochemistry, Graduate School of Medicine, Osaka Metropolitan University, Osaka, Japan, 5 Project for Prevention of HPV-related Cancer, Exploratory Oncology Research and Clinical Trial Center, National Cancer Center, Kashiwa, Chiba, Japan, 6 Spectrography and Bioimaging Facility, National Institute for Basic Biology, Okazaki, Aichi, Japan, 7 Natural Science Center for Basic Research and Development, Hiroshima University, Hiroshima, Japan, 8 Department of Clinical Neuroscience & Therapeutics, Graduate School of Biomedical and Health Sciences, Hiroshima University, Hiroshima, Japan

* mfukushi@hiroshima-u.ac.jp

**Data Availability Statement:** All relevant data are within the paper and its Supporting information files.

## Abstract

### Background

Optineurin (OPTN) is associated with several human diseases, including amyotrophic lateral sclerosis (ALS), and is involved in various cellular processes, including autophagy. Optineurin regulates the expression of interferon beta (IFNβ), which plays a central role in the innate immune response to viral infection. However, the role of optineurin in response to viral infection has not been fully clarified. It is known that optineurin-deficient cells produce more IFNβ than wild-type cells following viral infection. In this study, we investigate the reasons for, and effects of, IFNβ overproduction during optineurin deficiency both *in vitro* and *in vivo*.

### Methods

To investigate the mechanism of IFNβ overproduction, viral nucleic acids in infected cells were quantified by RT-qPCR and the autophagic activity of optineurin-deficient cells was determined to understand the basis for the intracellular accumulation of viral nucleic acids. Moreover, viral infection experiments using optineurin-disrupted (*Optn*-KO) animals were performed with several viruses.

### Results

IFNβ overproduction following viral infection was observed not only in several types of optineurin-deficient cell lines but also in *Optn*-KO mice and human ALS patient cells carrying

**Funding:** This study was supported by JSPS KAKENHI Grant Numbers 16K08812, 25460568, 26242085, 19K22968, 26830035 and 21K07461, and by the Tsuchiya Memorial Medical Foundation, the Program for Promotion of Basic and Applied Research for Innovations in BD (BRAIN), and the Takeda Science Foundation. This study was partially supported by the NIBB Individual Collaborative Research Program (ID: 12-361 and 12-340) and the Program of the Network-type Joint Usage/Research Center for Radiation Disaster Medical Science. The funders had no role in study design, data collection and analysis, decision to publish, or preparation of the manuscript.

**Competing interests:** The authors have declared that no competing interests exist.

mutations in *OPTN*. IFNβ overproduction in *Optn*-KO cells was revealed to be caused by excessive accumulation of viral nucleic acids, which was a consequence of reduced autophagic activity caused by the loss of optineurin. Additionally, IFNβ overproduction in *Optn*-KO mice suppressed viral proliferation, resulting in increased mouse survival following viral challenge.

## Conclusion

Our findings indicate that the combination of optineurin deficiency and viral infection leads to IFNβ overproduction *in vitro* and *in vivo*. The effects of optineurin deficiency are elicited by viral infection, therefore, viral infection may be implicated in the development of optineurin-related diseases.

## Introduction

*Optineurin* (*OPTN*) is a causative gene of several human diseases, including primary open-angle glaucoma, Paget's disease of bone, Crohn's disease, and amyotrophic lateral sclerosis (ALS) [1]. Optineurin is ubiquitously expressed and is a homologue of nuclear factor-kappa B (NF-κB) essential modulator (NEMO) [2]. Optineurin consists of several domains, including two coiled-coil domains, a leucine zipper, an LC3-interacting region, the ubiquitin-binding domain of ABIN protein and NEMO (UBAN), and a zinc finger domain, which mediate binding with many cellular proteins, such as TANK-binding kinase 1 (TBK1), and some viral proteins [3]. Optineurin serves multiple roles in several cellular processes, including NF-κB signalling, through interaction with many binding partners [2].

Optineurin inhibits the transcription factors, NF-κB and interferon regulatory factor 3 (IRF3), both of which bind to the promoter of *interferon beta* (*IFNB*) and induce its expression [4–7]. IFNβ is immediately and transiently secreted from almost all cell types in response to viral infections and leads an effective immune response that eliminates the invading virus [8]. Secreted IFNβ acts in both an autocrine and paracrine manner, which produces the dual effects of inducing apoptosis of infected cells and triggering an anti-viral state in uninfected cells. In addition, IFNβ promotes further immune responses, including inflammation, by the induction of hundreds of downstream genes [9]. IFNβ also enhances optineurin expression following viral infection, whereas optineurin suppresses IFNβ production, suggesting that optineurin negatively regulates IFNβ production following viral infection [10]. Furthermore, during viral infection, optineurin-defective cells produce an excess of IFNβ as compared with wild-type (WT) cells [10–12]. Therefore, the relationship between optineurin and IFNβ is thought to play an important role in viral infection.

Optineurin functions as an autophagy receptor and optineurin deficiency therefore reduces autophagosome formation during both basal and starvation-induced autophagy [13, 14]. Autophagy was originally characterised as an intracellular recycling system that maintains cell homeostasis. However, it is now also recognised as a process for the elimination of unwanted substances and microorganisms from cells [15]. Indeed, autophagy is involved in multiple diseases, including infectious diseases and neurodegenerative disorders [16]. During bacterial infection, optineurin cooperates with other autophagy receptors, sequestosome 1 and nuclear dot protein 52 kDa (NDP52), to capture invading pathogens [13, 17]. The captured bacteria are sequestered from the cytosol in an autophagosome, which then merges with a lysosome to

digest the bacteria [2]. By contrast, the roles of optineurin in autophagy during viral infection are not well understood.

We have found *OPTN* mutations in several ALS patients [4]. These mutations result in loss of the optineurin protein itself or loss of its function. Therefore, in this study, we investigated the influence of viral infection on IFNβ production *in vitro* and *in vivo* under optineurin-defective conditions. We demonstrated that a combination of defective optineurin and viral infection leads to excess production of IFNβ in cells and mouse lungs, resulting in increased survival rates of animals following viral infection. We also elucidated the mechanism by which reduced autophagic activity caused by optineurin deficiency leads to IFNβ overproduction. IFNβ can induce many downstream genes; therefore, there might be a connection between viral infection and optineurin-related diseases.

## Materials and methods

### Cells

HEK293 human embryonic kidney fibroblasts, J774.1 mouse macrophages, BV-2 mouse microglial cells, and *Sqstm1*-KO mouse embryonic fibroblasts (MEFs) [18] were maintained in Dulbecco's modified Eagle's medium (DMEM) or RPMI 1640 medium supplemented with 10% foetal calf serum (FCS), 50 U/ml penicillin, and 50 μg/ml streptomycin, at 37˚C with 5% $CO_2$. *Optn*-KO MEFs were isolated from mice, immortalised by introduction of simian virus 40 large T-antigen, and maintained in DMEM with 10% FCS, penicillin, and streptomycin, at 37˚C with 5% $CO_2$. Primary medaka (*Oryzias latipes*) cells were isolated from fish and maintained in L-15 medium supplemented with 20% FCS, 50 U/ml penicillin, and 50 μg/ml streptomycin, at 30˚C without $CO_2$. *Optn*-KO clones of GFP-LC3-RFP-LC3ΔG MEFs [19] were established using the CRISPR/Cas9 technique as follows. Briefly, GFP-LC3-RFP-LC3ΔG MEFs were transfected with recombinant Cas9 protein and a guide RNA that targeted exon 2 of the mouse *Optn* gene. Forty-eight hours after transfection, cells were seeded on 96-well plates at a low concentration for cell cloning. Disruption of the *Optn* gene and loss of optineurin protein in cell clones were confirmed by sequencing and western blotting, respectively. Accordingly, two independent *Optn*-KO clones were established. J774.1 and BV-2 cells with low levels of optineurin protein and control cells were established by infection with a lentivirus that carried a small hairpin (sh) RNA to target the *Optn* gene or scrambled RNA, respectively. Primary astrocytes were isolated from mouse pups and were stained with anti-GFAP, anti-Iba1 and anti-neurofilament H antibodies to confirm purity. Human fibroblasts were isolated from ALS patients carrying the optineurin mutations, p.Q398* or p.E478G, and from healthy donors. Cells from an ALS-optineurin:p.Q398* patient and a healthy donor were immortalised by introduction of simian virus 40 large T-antigen and human telomerase reverse transcriptase (TERT) using a lentivirus. Cells from an ALS-optineurin:p.E478G patient and a healthy donor were immortalised by lentivirus-edited gene transfer of a human cyclin-dependent kinase 4 mutant (R24C) and TERT [20]. These established human cells were maintained in DMEM with 10% FCS, penicillin, and streptomycin. ALS patient fibroblasts with causative mutations in *SOD1* (identification numbers: ND29149 and ND39026), *FUS* (ND29563 and ND39027), *ANG* (ND29689), *FIG4* (ND39025), *TARDBP* (ND41003, this gene encodes TDP-43), and healthy donor fibroblasts (ND29510) were provided by the Coriell Institute through the National Institute of Neurological Disorders and Stroke Repository. *OPTN* and/or *CAL-COCO2* (which encodes NDP52)-disrupted HeLa cells were established by the CRISPR/Cas9 technique and were maintained in DMEM with 10% FCS, penicillin, and streptomycin.

## Viruses, antibodies, and reagents

Sendai virus (SeV) Cantell and Z strains were grown in chicken eggs 11 days after fertilisation. Influenza virus PR8 strain was grown in the lungs of BALB/c mice. Betanodavirus was used for medaka infection experiments. Titres of SeV and influenza viruses were measured by infecting macaque monkey kidney-derived LLC-MK2 cells, and Madin–Darby canine kidney cells, respectively. Anti-optineurin polyclonal (Cayman, #100000), anti-optineurin monoclonal (Santa Cruz Biotechnology, sc-166576), anti-LC3 monoclonal (MBL, M186-3), anti-GFAP polyclonal (Abcam, ab7260), anti-Iba1 polyclonal (Wako, #019–19741), anti-neurofilament monoclonal (Covance, SMI32), anti-actin monoclonal (Chemicon, MAB1501), horseradish peroxidase (HRP)-conjugated donkey anti-mouse IgG polyclonal (Abcam, ab98799), and HRP-conjugated donkey anti-rabbit IgG polyclonal (Abcam, ab97085) antibodies were used for western blotting and immunocytochemistry. The autophagy inhibitor, wortmannin (Invi-voGen), was added to culture media at the indicated concentrations. This study was approved by the Hiroshima University biosafety committee for living modified organisms (approved number: 2022-316-2).

## Animals

*Optn*-KO mice on the C57BL/6 background were generated as described previously [21]. The mice were bred in-house and used under specific pathogen-free conditions. Eight-to-nine-week-old *Optn*-KO mice were used in the viral challenge assay. Pathogen-free C57BL/6 mice of the same age and sex were purchased from Charles River Laboratories Japan as WT controls. *Optn*-KO medaka carrying the optineurin mutations p.Q64* were generated by the targeting induced local lesions in genome (TILLING) method using N-ethyl N-nitrosourea and were screened by high resolution melting curve analysis [22, 23]. Breeding and infection experiments for all mice were performed by operators who trained animal care and handling in the restricted area with biological safety level 3 (BSL-3) for infection experiment at the Animal Care Unit of Hiroshima University in accordance with the guidelines of the Hiroshima University Animal Research Committee (approval number: A15-133).

## Viral infection of cells and animals

Cells were seeded one day before viral infection. Cells were inoculated with virus at a multiplicity of infection of 20 and incubated for 1 hour at 37˚C. To examine the susceptibility of cells to virus, incubation with virus was performed at 4˚C to monitor virus attachment. After discarding free virus, culture medium was added to the cells. Culture supernatants and cell lysates at the indicated times were used for analysis. To examine the relationship between autophagic activity and viral defective interfering (DI) genome quantity, WT MEFs were inoculated with SeV (Cantell strain) and cultured in medium containing wortmannin or DMSO at the indicated concentrations. After incubation for 1.5 hours, the cells were harvested and used for qPCR analysis. In mouse experiments, mice were anesthetised with isoflurane and intranasally inoculated with SeV [Z strain, $3 \times 10^5$ cell infectious units (CIU)/mouse] or influenza virus [15 plaque-forming units (PFU)/mouse] [24]. The infected mice were observed daily for signs of illness, measured their body weights and monitored for 20 days to assess the survival rate. Mice that lost more than 25% of their original body weight were immediately euthanized using isoflurane for humane reasons, and that day was used as the humane endpoint for assessing survival. No mice died before meeting criteria for euthanasia. For IFNβ production and determination of viral titre in mouse lungs, mice were infected with SeV (Cantell strain, $2 \times 10^8$ CIU/mouse; Z strain, $3 \times 10^5$ CIU/mouse). IFNβ production was quantified by enzyme-linked immunosorbent assays (ELISAs) at the indicated times after infection. Mouse

lungs were excised, homogenised in 1 ml physiological saline using metal beads, and clarified by centrifugation. The clarified supernatants containing virus were used for qPCR and plaque-forming assays to measure viral titres. Medaka were virally challenged by being maintained in water that contained betanodavirus ($2 \times 10^5$ 50% tissue culture infective dose/ml) and were monitored for 10 days. Some experiments in this study were carried out at the Natural Science Center for Basic Research and Development in Hiroshima University.

## Luciferase assay

HEK293, MEFs, and ALS patient cells carrying the optineurin:p.Q398* variation were transfected with an optineurin-expression vector. An empty vector was transfected as a control. Separate luciferase reporter plasmids containing binding sites for NF-κB, IRF3, or activator protein 1 (AP-1), plus the *IFNB* promoter, and the interferon stimulation response element (*ISRE*) (Promega), were also transfected into the cells. Transfected cells were infected on the following day with SeV (Cantell strain), and 24 hours later, the infected cells were lysed in lysis buffer. Luciferase activity in the cell lysates was measured using a Berthold Tech TriStar luminometer and MikroWin software (Version 4.41). Relative luciferase units are indicated as RLU in figures.

## ELISA

Culture media of cells infected with virus were collected at the indicated times and used for ELISA quantification of IFNβ (R&D Systems) in accordance with the manufacturer's instructions. Mouse lungs were excised, homogenised in 1 ml physiological saline using metal beads, and clarified by centrifugation. The clarified supernatants were used for IFNβ ELISA. Absorbance was measured using a Berthold Tech TriStar luminometer and MikroWin software (Version 4.41).

## Quantitative PCR (qPCR)

To measure the viral titre in infected mouse lungs and the copy number of the viral genome, total RNA was isolated from clarified lung homogenates and cultured cells using RNeasy (Qiagen). The RT reaction and qPCR were performed using ReverTra Ace and Thunderbird SYBR (Toyobo), respectively, in accordance with the manufacturer's instructions. The primer mix provided in ReverTra Ace was used for the RT reaction. Primer sets for qPCR were 5′–GGACAAGTCCAAGACTTCCAG–3′ and 5′–GCCAGGATTCCCGTTGAATA–3′ for the SeV DI genome, 5′–CTGACAACACAGACTCCCTTAC–3′ and 5′–GGTCTCCATAGATGGGTCAAAC–3′ for the SeV full genome, and 5′–CAGCCTTCCTTCTTGGGTATG–3′ and 5′–GGCATAGAGGTCTTTACGGATG–3′ for mouse β-actin. Applied Biosystems StepOnePlus Real-Time PCR system (Thermo Fisher Scientific) and its software were used for qPCR and collecting and analysing data in accordance with the manufacturer's instructions.

## Plaque-forming assay

A plaque-forming assay was performed to measure the titre of influenza virus in the lungs of infected mice [25]. In brief, clarified lung homogenates that contained the virus were serially diluted in MEM containing 0.2% bovine serum albumin, 2 mM L-glutamine, 50 U/ml penicillin, and 50 mg/ml streptomycin. Viral dilutions were used to infect Madin–Darby canine kidney cell monolayers for 1 hour at 37˚C. Cells were washed once with phosphate-buffered saline to remove free viruses, overlaid with modified MEM containing 0.6% agar, 0.2% bovine serum albumin, 0.01% DEAE-dextran, 25 mM HEPES, and 1 mg/ml trypsin, and incubated at

37˚C. After incubation for 2 days, the monolayer cells were stained with a crystal violet solution (0.095% crystal violet and 19% methanol). After washing with tap water, plaques were counted.

## Flow cytometry

To measure the clearance of exogenous nucleic acids from the cytoplasm, MEFs were transfected with poly(I:C)-rhodamine (InvivoGen) using transfection reagent LyoVec (InvivoGen) for 1 hour. The cells were washed with phosphate-buffered saline to remove free poly(I:C)-rhodamine and then culture medium was added. After incubation for 0, 3, and 6 hours, the cells were detached, washed with phosphate-buffered saline, and analysed using an LSRFortessa X-20 flow cytometer (BD) to measure rhodamine signal intensity. To measure the quantity of autophagosomes, a CYTO-ID autophagy detection kit 2.0 (Enzo) was used. MEFs in a steady state, starved state (DMEM without amino acids), or SeV Cantell-infected state were stained with the CYTO-ID reagent as per the manufacturer's instructions. The green fluorescent protein (GFP) signal from cells was measured using a FACSVerse (BD). To measure autophagic flux, GFP and red fluorescent protein (RFP) fluorescence intensities of parental GFP-LC3-RFP-LC3ΔG MEFs and *Optn*-KO clones infected with SeV (Cantell strain) or mock infected were measured using an LSRFortessa X-20 flow cytometer. Next, the GFP/RFP ratio was calculated using the median GFP and RFP signal intensities in each sample. Autophagy flux of infection was calculated as a decreased GFP/RFP ratio of virus-infected compared with mock samples.

## Microscopy

MEFs stained with CYTO-ID reagent and mouse astrocytes stained with antibodies were observed under an ECLIPSE TE2000-S microscope (Nikon) and Zeiss Axio Observer Z1 with ZEN Imaging Software (Carl Zeiss).

## Antibody array

Multiple human and mouse cytokines were analysed using Proteome Profiler Antibody Arrays (R&D Systems, ARY005 and ARY006). The kit membranes were incubated with culture media from human fibroblasts and mouse astrocytes infected with SeV or mock infected in accordance with the manufacturer's instructions.

## Statistical analysis

All data are presented as mean values ± SD. BellCurve for Excel software (Social Survey Research Information Co., Ltd.) was used for all statistical evaluations. Statistical significance was determined using two-tailed unpaired Student's *t*-test, two-tailed unpaired Welch's *t*-test, one-way ANOVA followed by Dunnett's test, and two-way ANOVA followed by the Tukey–Kramer method. The statistical significance of the survival rates of mice and medaka was determined by the Kaplan–Meier method. A p-value of less than 0.05 was considered statistically significant. *$p < 0.05$, **$p < 0.01$, ***$p < 0.001$.

# Results

## IFNβ overproduction by optineurin-defective cells following viral infection is caused by low clearance of viral nucleic acids

We investigated a possible interaction between optineurin and viral infection in IFNβ production because we previously showed that optineurin inhibits the activities of NF-κB and IRF3

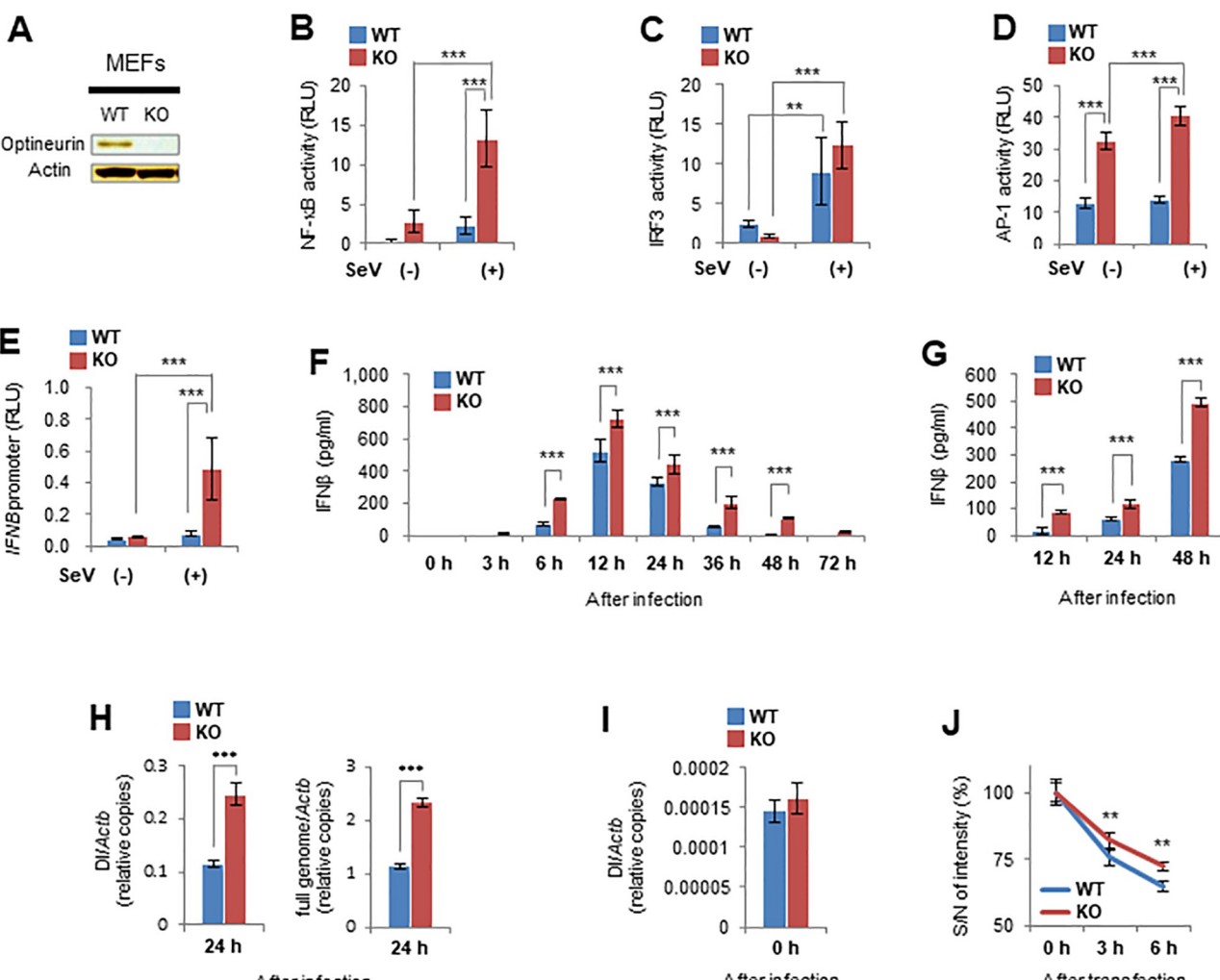

**Fig 1. IFNβ overproduction in *Optn*-KO MEFs following viral infection is caused by accumulation of viral RNA.** (A) Optineurin protein levels in WT and *Optn*-KO MEFs examined by western blotting. (B–E) Transcriptional activity of NF-κB (*n* = 5 each), IRF3 (*n* = 5 each), AP-1 (*n* = 5 each), and *IFNB* promoter (*n* = 5 each) in WT and *Optn*-KO MEFs infected with SeV (Cantell strain) was measured by luciferase assays at 24 hours after inoculation. (F and G) IFNβ production by WT and *Optn*-KO MEFs infected with SeV (Cantell or Z strains) was measured by ELISA at the indicated hours after inoculation. *n* = 3 independent replicates for each indicated time. (H) Relative copy numbers of viral DI and the full genomes in WT (*n* = 4) and *Optn*-KO (*n* = 4) MEFs infected with SeV (Cantell strain) were measured by quantitative PCR (qPCR) at 24 hours after inoculation. (I) Relative viral DI genome copy numbers in WT (*n* = 3) and *Optn*-KO (*n* = 3) MEFs infected with SeV (Cantell strain) were measured by qPCR immediately after inoculation. Viral inoculation was performed at 4°C to monitor virus attachment. (J) Fluorescence intensities of WT (*n* = 4) and *Optn*-KO (*n* = 4) MEFs transfected with rhodamine-labelled poly(I:C) were measured by flow cytometry at the indicated times after transfection. The ratio of signal and noise intensities was then calculated. Data are presented as mean values ± SD. Two-way ANOVA followed by the Tukey–Kramer method (B–G) and two-tailed unpaired Student's *t*-test (H–J) were applied for statistical analyses. **$p < 0.01$, ***$p < 0.001$.

[4, 6]. Overexpression of optineurin in HEK293 cells infected with the Cantell strain of SeV inhibited the transcriptional activities of NF-κB, IRF3, and the *IFNB* promoter as previously reported (S1A–S1C Fig) [10, 11]. Next, to examine IFNβ production induced by viral infection in the absence of optineurin, we established MEFs from *Optn*-KO mice (Fig 1A). They showed no difference in morphology, cell growth, or viability compared with WT MEFs (S1D–S1F Fig). The activities of NF-κB, AP-1, and the *IFNB* promoter in *Optn*-KO MEFs were greater than those in WT MEFs after viral infection (Fig 1B, 1D and 1E). The transcriptional activities of IRF3 in both *Optn*-KO and WT MEFs were greater by viral infection (Fig 1C). The secretion

of IFNβ from *Optn*-KO MEFs was also greater than that from WT MEFs after viral infection with Cantell or Z strains of SeV (Fig 1F and 1G). Although IFNβ is generally produced from the host cell in response to viral infection, SeV Cantell strain is known to strongly induce IFNβ production because of its RNA-based DI genome, which is not present in Z strain [26]. Therefore, to elucidate the reason for IFNβ overproduction in virus-infected *Optn*-KO cells, we quantified the viral DI genome copy number in SeV Cantell strain-infected cells. Viral DI genome quantity in *Optn*-KO MEFs was higher compared with that in WT cells at 6 and 24 hours after infection, as was the full viral genome (Fig 1H and S1G Fig). There was no difference in viral DI genome quantity between *Optn*-KO and WT MEFs immediately after infection, which indicates that the viral susceptibilities of *Optn*-KO and WT MEFs were similar (Fig 1I). Next, to measure the ability of *Optn*-KO MEFs to clear extrinsic nucleic acids, fluorescent dye-labelled poly(I:C) was transfected into MEFs. Signal intensities of *Optn*-KO MEFs were stronger than those of WT MEFs at 3 and 6 hours after transfection, which indicates that *Optn*-KO MEFs fail to eliminate exogenous nucleic acids efficiently (Fig 1J). These results show that accumulation of viral DI genomes in infected *Optn*-KO cells is responsible for IFNβ overproduction.

## Optineurin deficiency causes low autophagic activity

To clarify the reason for the accumulation of DI genomes in virus-infected *Optn*-KO MEFs, we examined autophagic activity of *Optn*-KO cells because optineurin is an autophagy receptor. Autophagosomes, structures in the sequential digestive process of autophagy, were stained using the specific marker, CYTO-ID. Signal intensities of autophagosomes in *Optn*-KO MEFs were weaker than those in WT cells in virus-infected and steady states (Fig 2A and 2B). Moreover, conversion of LC3-I to LC3-II in *Optn*-KO MEFs was less than that in WT MEFs (Fig 2C). Differences between *Optn*-KO and WT MEFs in autophagosome signal intensities and LC3 conversion in a starved state, which is a representative autophagic stimulus, were similar to those in steady and virus-infected states (S2A and S2B Fig). These results indicate that the autophagic activity of *Optn*-KO MEFs is always weaker than that of WT MEFs with or without stimulation, such as by viral infection or starvation. To further examine whether an optineurin defect reduces autophagic activity during viral infection, we disrupted *Optn* using the CRISPR/Cas9 technique in the GFP-LC3-RFP-LC3ΔG MEF cell line, in which autophagic activity is easily measured (S2C Fig). As expected, two independent *Optn*-KO clones produced significantly greater quantities of IFNβ compared with parental cells after viral infection (Fig 2D). The ratio of autophagy flux in the virus-infected state to that in the steady state was smaller in both *Optn*-KO clones than in the parental control (Fig 2E). These results show that the autophagic activity of *Optn*-KO cells is weaker than that of WT cells. Next, to examine whether the reduced autophagic activity is involved in viral DI genome accumulation, we quantified the viral DI genome in the presence of the autophagy inhibitor, wortmannin. There was a larger amount of viral DI genome in infected cells treated with wortmannin than in cells treated with DMSO (Fig 2F). From these results, we concluded that the reduced autophagic activity in *Optn*-KO cells leads to incomplete clearance and accumulation of viral genomes during infection, which results in IFNβ overproduction.

## IFNβ overproduction in response to viral infection is common in other cell types with optineurin deficiency

To examine IFNβ overproduction in optineurin-defective cells other than MEFs, we established *Optn*-knockdown (KD) J774.1 macrophages and BV-2 microglial cells (Fig 3A). IFNβ production by *Optn*-KD J774.1 and BV-2 cells was greater than by control cells during viral

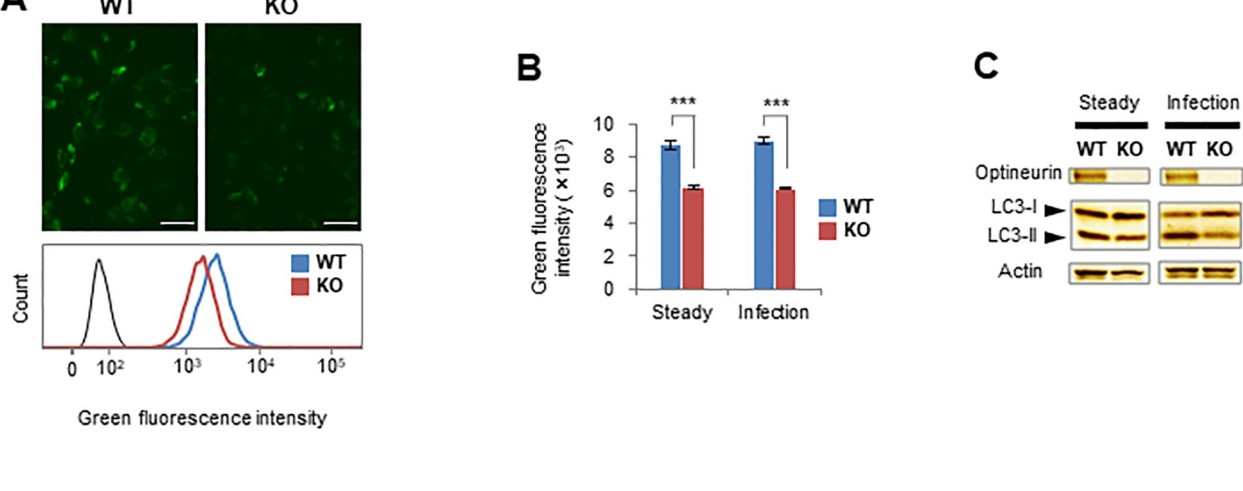

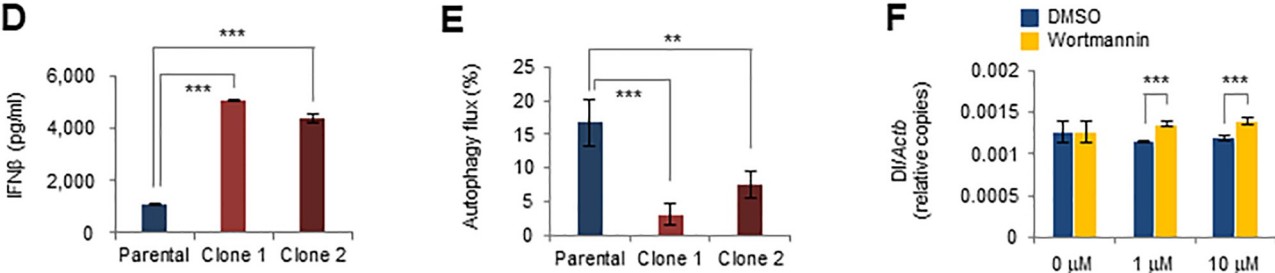

**Fig 2. Defective optineurin causes autophagy failure.** (A) (upper panels) Fluorescence images of WT and *Optn*-KO MEFs stained with the autophagy marker, CYTO-ID. Scale bars, 50 μm. (lower panel) Fluorescence intensities of WT and *Optn*-KO MEFs stained with CYTO-ID were measured by flow cytometry and the results were compared. The black line indicates the unstained control. (B) Fluorescence intensities of WT ($n = 4$) and *Optn*-KO ($n = 4$) MEFs in steady and virus-infected states were measured by flow cytometry and the averages of the median fluorescence intensities are indicated. (C) LC3-I, LC3-II, optineurin, and actin of WT and *Optn*-KO MEFs in steady and virus-infected states were examined by western blotting. (D) IFNβ production by parental ($n = 4$) and two *Optn*-KO clones ($n = 4$ each) of GFP-LC3-RFP-LC3ΔG MEFs was measured by ELISA at 14 hours after inoculation. (E) Autophagy flux after infection is shown as a decreased GFP/RFP ratio in parental ($n = 4$) and two *Optn*-KO clones ($n = 4$ each) of GFP-LC3-RFP-LC3ΔG MEFs infected with SeV (Cantell strain). The GFP and RFP signal intensities in parental and two *Optn*-KO clones of GFP-LC3-RFP-LC3ΔG MEFs infected with SeV (Cantell strain) or mock treated were measured by flow cytometry. The GFP/RFP ratios were calculated using the median GFP and RFP signal intensities in each sample. The decreased ratios were calculated by comparing GFP/RFP ratios of virus-infected and mock samples. (F) Relative viral DI genome copy numbers in SeV-infected WT MEFs treated with wortmannin ($n = 3$) or DMSO ($n = 3$) at the indicated concentrations for 1.5 hours were measured by qPCR. Data are presented as mean values ± SD. Two-way ANOVA followed by the Tukey–Kramer method (B and F) and one-way ANOVA followed by Dunnett's test (D and E) were applied for statistical analyses. **p < 0.01, ***p < 0.001.

infection (Fig 3B and 3C). Additionally, IFNβ production by primary astrocytes isolated from *Optn*-KO mouse pups was greater after viral infection (Fig 3D and S3 Fig). These results show that optineurin-defective immune and neural cells overproduced IFNβ following viral infection, in a similar manner to *Optn*-KO fibroblasts.

## IFNβ production by ALS patient cells and autophagy receptor-disrupted cells during viral infection

We examined IFNβ production by ALS patient cells that carried the optineurin amino acid substitution mutation, p.Q398* or p.E478G, which we described previously [4]. *IFNB* promoter activity and IFNβ production by ALS patient cells with either optineurin mutant were significantly greater than in healthy donor cells (Fig 4A, S4A and S4B Fig). Next, to confirm

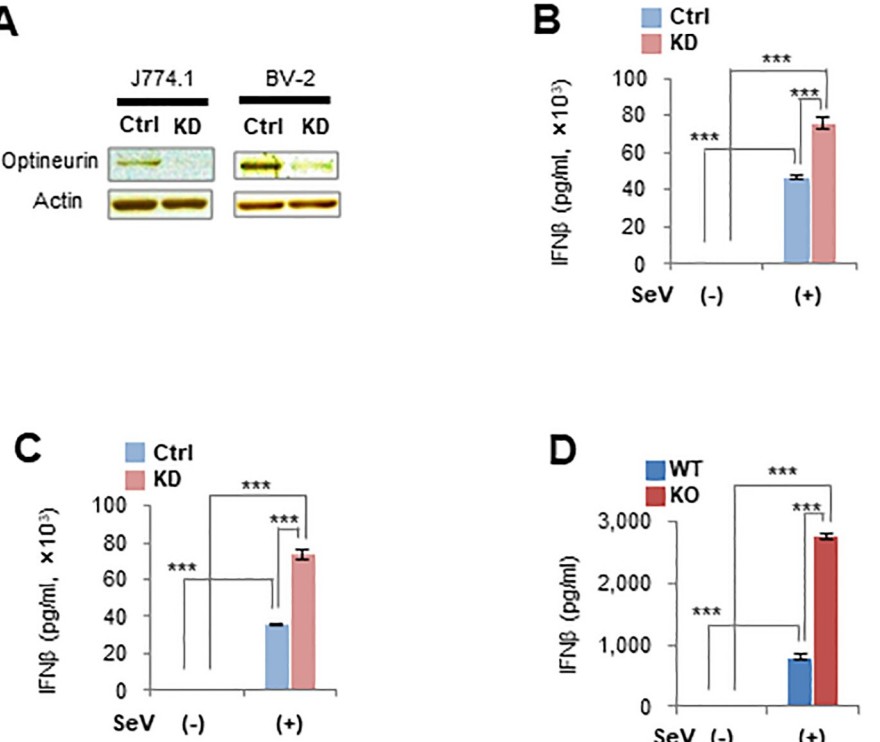

**Fig 3. IFNβ overproduction following viral infection is common in other cell types with optineurin deficiency.** (A) Optineurin protein levels in scramble RNA-introduced (Ctrl) and *Optn*-KD J774.1 mouse macrophage and BV-2 mouse microglial cells were examined by western blotting. (B) IFNβ production by Ctrl ($n = 4$) and *Optn*-KD ($n = 4$) J774.1 macrophages infected with SeV (Cantell strain) was measured by ELISA at 6 hours after inoculation. (C) IFNβ production by Ctrl ($n = 4$) and *Optn*-KD ($n = 4$) BV-2 microglial cells infected with SeV (Cantell strain) was measured by ELISA at 6 hours after inoculation. (D) IFNβ production by WT ($n = 4$) and *Optn*-KO ($n = 4$) primary astrocytes infected with SeV (Cantell strain) was measured by ELISA at 12 hours after inoculation. Data are presented as mean values ± SD. Two-way ANOVA followed by the Tukey–Kramer method (B–D) were applied for statistical analyses. ***$p < 0.001$.

whether IFNβ overproduction during viral infection is a common phenomenon in all ALS patients, we examined IFNβ production by cells from several ALS patients with mutations in the ALS-causative genes, *SOD1*, *FUS*, *ANG*, *FIG4*, and *TARDBP*. IFNβ overproduction was not observed in these cells (S4B Fig). Next, because optineurin is an autophagy receptor, we examined IFNβ production by cells with disruption to other autophagy receptor genes, *Sqstm1* and *CALCOCO2*, which encode sequestosome 1 and NDP52, respectively. *Sqstm1*(+/−) MEFs produced a significantly larger amount of IFNβ compared with *Sqstm1*(+/+) and *Sqstm1*(−/−) MEFs after viral infection (Fig 4C). *CALCOCO2*-, *OPTN*-, and *CALCOCO2/OPTN*-disrupted HeLa cells also produced larger amounts of IFNβ compared with parental cells after viral infection (Fig 4D). These results show that IFNβ overproduction caused by a combination of optineurin deficiency and viral infection occurs in humans as well as in mice. Moreover, IFNβ overproduction following viral infection was common to autophagy receptor-defective cells but not to cells with mutations in other ALS-causative genes.

## Effect of optineurin deficiency during viral infection *in vivo*

To clarify the *in vivo* effect of optineurin deficiency, we infected *Optn*-KO and WT mice with viruses. IFNβ production in the lungs of *Optn*-KO mice was greater compared with

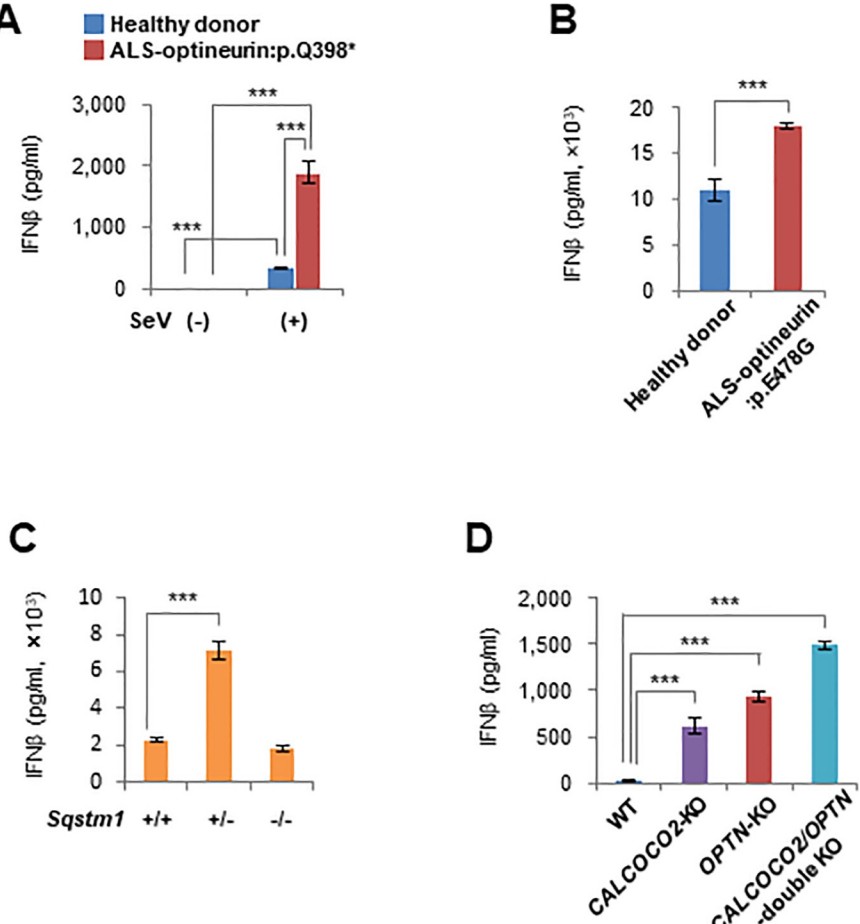

**Fig 4. IFNβ production by ALS patient cells and autophagy receptor-disrupted cells during viral infection.** (A) IFNβ production by healthy donor ($n = 4$) and ALS-optineurin:p.Q398* patient ($n = 4$) fibroblasts infected with SeV (Cantell strain) was measured by ELISA at 24 hours after inoculation. (B) IFNβ production from healthy donor ($n = 4$) and ALS-optineurin:p.E478G patient ($n = 4$) fibroblasts infected with SeV (Cantell strain) was measured by ELISA at 24 hours after inoculation. (C) IFNβ production by *Sqstm1*(+/+), *Sqstm1*(+/−) and *Sqstm1*(−/−) MEFs ($n = 4$ each) infected with SeV (Cantell strain) was measured by ELISA at 12 hours after inoculation. (D) IFNβ production by *CALCOCO2*- and/or *OPTN*-disrupted HeLa cells ($n = 3$ each) infected with SeV (Cantell strain) was measured by ELISA at 24 hours after inoculation. Data are presented as mean values ± SD. Two-way ANOVA followed by the Tukey–Kramer method (A), two-tailed unpaired Student's *t*-test (B), and one-way ANOVA followed by Dunnett's test (C and D) were applied for statistical analyses. ***p < 0.001.

that in WT mice after infection with SeV Cantell strain or Z strain (Fig 5A and 5B). Mouse survival rates were examined using the SeV Z strain and influenza virus because all of the *Optn*-KO and WT mice infected with the maximum titre of the SeV Cantell strain survived. Survival rates of *Optn*-KO mice were significantly higher than those of WT mice after infection with the SeV Z strain and influenza virus (Fig 5C and 5D). Consistently, viral titres in the lungs of *Optn*-KO mice were lower than those of WT mice (Fig 5E and 5F). In addition, the survival rate of *Optn*-KO medaka was higher than that of WT medaka after infection with a fish virus (S5A and S5B Fig). These results demonstrate that optineurin deficiency leads to IFNβ overproduction during viral infection *in vivo*, similar to the response *in vitro*, and results in suppressed mortality from viral infection.

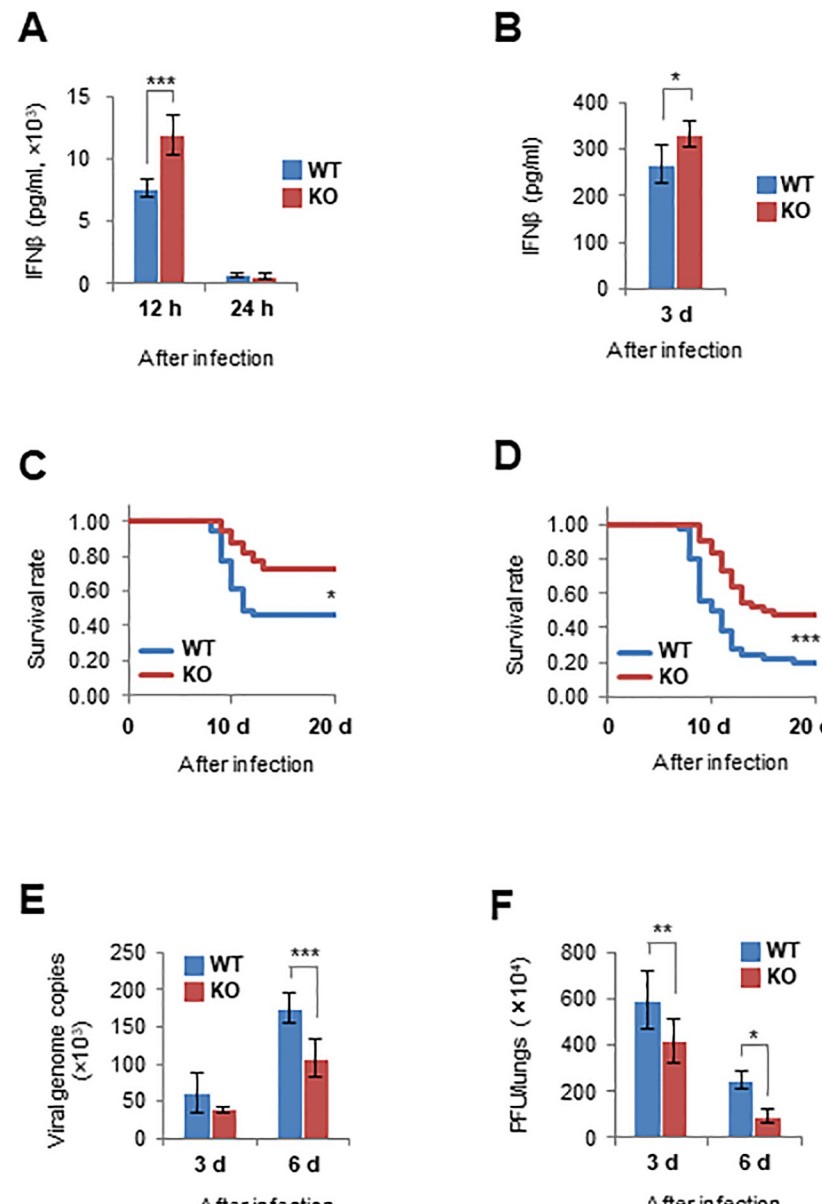

**Fig 5. Optineurin is involved in IFNβ production following viral infection *in vivo*.** (A) IFNβ production in the lungs of WT and *Optn*-KO mice infected with SeV (Cantell strain) was measured by ELISA at the indicated hours after inoculation. At 12 h after infection, $n = 4$ and 3 for WT and *Optn*-KO mice, respectively. At 24 h after infection, $n = 5$ and 4 for WT and *Optn*-KO mice, respectively. (B) IFNβ production in the lungs of WT ($n = 4$) and *Optn*-KO ($n = 4$) mice infected with SeV (Z strain) was measured by ELISA at 3 days after inoculation. (C) Survival rates of WT ($n = 39$) and *Optn*-KO ($n = 40$) mice infected with SeV (Z strain). (D) Survival rates of WT ($n = 50$) and *Optn*-KO ($n = 44$) mice infected with influenza virus (PR8 strain). (E) Viral genome copy numbers of SeV (Z strain) in the lungs of WT (3d, $n = 6$; 6d, $n = 7$) and *Optn*-KO (3d, $n = 6$; 6d, $n = 8$) mice were measured by quantitative PCR to determine viral titres. (F) Influenza virus titres in the lungs of WT (3d, $n = 5$; 6d, $n = 4$) and *Optn*-KO (3d, $n = 5$; 6d, $n = 4$) mice were measured by plaque-forming assays. Data are presented as mean values ± SD. Two-way ANOVA followed by the Tukey–Kramer method (A, E and F), two-tailed unpaired Student's *t*-test (B), and Kaplan–Meier method (C and D) were applied for statistical analyses. *p < 0.05, **p < 0.01, ***p < 0.001.

## Discussion

In this study, we demonstrated that a combination of optineurin deficiency and viral infection induces the overproduction of IFNβ *in vitro* and *in vivo* compared with WT controls. We also elucidated that this IFNβ overproduction is caused by accumulation of viral RNA as a consequence of low autophagic activity. In addition, the survival rate of *Optn*-KO animals was higher than that of WT animals in viral challenges.

Our *in vitro* study showed that IFNβ overproduction is induced by viral infection in optineurin-deficient cells, as has been shown previously [10–12]. IFNβ overproduction following viral infection was observed in several cell types, such as optineurin-deficient macrophages, microglial cells, and astrocytes and also in human ALS patient cells carrying optineurin mutations. We also observed IFNβ overproduction in the lungs of virus-infected *Optn*-KO mice. Therefore, the phenomenon of IFNβ overproduction in response to combined optineurin defects and viral infection is common regardless of cell type and *in vitro/in vivo* conditions. However, regarding IFNβ production in optineurin-deficient cells, there are discrepancies between different reports [10, 11, 27–30]. We attribute these discrepancies to the cell types, stimuli, and methods used in the studies. In brief, some groups used bone marrow-derived macrophages (BMDM), LPS and poly(I:C). LPS and poly(I:C) were directly added to the culture media of BMDM, which express Toll-like receptors (TLRs). Therefore, the stimuli of LPS and poly(I:C) are thought to be transmitted into cells through TLR 4 located on the cell surface or TLR 3 located in endosome. In our study, fibroblasts were mainly used that do not express TLRs, and viral infection was used as a stimulus. Viruses directly enter into the cytoplasm, and their genomes are recognised by cytoplasmic viral RNA sensors, such as retinoic acid-inducible gene I (RIG-I) and melanoma differentiation-associated protein 5 (MDA5), not by TLRs. Other groups also reported excess IFNβ production in optineurin-deficient fibroblasts during viral infection [10, 11]. On the basis of these findings, we propose that the discrepancy in IFNβ production in optineurin-deficient cells is caused by a difference in the signalling pathways induced in response to stimuli.

*Optn*-KO mice and medaka had high survival rates compared with WT animals in our viral challenge. However, these mouse results were opposite to that of a previous study [29]. Here, we demonstrated low viral titres in the lungs of *Optn*-KO mice compared with WT mice after challenge with SeV or influenza virus, which both cause viral pneumonia [24, 31]. We also revealed IFNβ overproduction in the lungs of virus-infected *Optn*-KO mice, similar to the results of the *in vitro* study using optineurin-deficient cells. IFNβ strongly inhibits viral proliferation; therefore, the low viral titres observed in the lungs of mice are reasonable. Generally, virus actively proliferates in infected lungs in viral pneumonia and the amount of virus is well correlated with the degree of pneumonia [24]. In influenza, anti-viral drugs can be administered to inhibit viral proliferation and improve pneumonia [25]. This explains the low viral titres and the high survival rates of *Optn*-KO mice. Based on the results of our mouse experiments, we suggest that the high survival rate of *Optn*-KO medaka is because of IFNβ overproduction.

IFNβ is produced immediately, but only transiently, in response to viral infection and leads to an effective immune response that eliminates invading pathogens [8]. Although IFNβ induces hundreds of downstream genes involved in the immune response, the amount and duration of IFNβ production is strictly regulated to prevent an excessive immune response. Secreted IFNβ acts in both an autocrine and paracrine manner, which produces the dual effects of inducing apoptosis in infected cells and triggering an anti-viral state in uninfected cells [8, 32]. In fact, *Optn*-KO MEFs, which can produce IFNβ excessively, showed increased levels of cell death after viral infection than WT cells (S6 Fig). From these facts, we speculate

that the elimination of the virus by the rapid death of infected cells and the inhibition of viral replication by an anti-viral state in uninfected cells strongly suppresses the spread of the virus in *Optn*-KO animals. Consequently, the inhibition of viral spread leads to the improved survival of infected *Optn*-KO animals.

The relationship between IFNβ and optineurin is thought to form a negative feedback loop, in which IFNβ stimulates optineurin expression and optineurin inhibits IFNβ expression [10]. The physiological significance of this negative feedback has been unclear. However, we observed here that virus-infected optineurin-deficient cells secrete excessive amounts of not only IFNβ but also several inflammatory cytokines, some of which are induced by IFNβ (S7 Fig). Considered together, this negative feedback may therefore play a role in preventing harmful hyperinflammation. IFNβ plays a pivotal role in innate immunity against viral infection; it is a strong anti-viral agent itself and induces numerous cytokines that protect the host against pathogen invasion. Therefore, the spatial-temporal levels of IFNβ need to be strictly regulated. For example, IFNβ overproduction is responsible for the type I interferonopathy genetic disorders, Aicardi–Goutières syndrome and Singleton–Merten syndrome, which are characterised by several symptoms associated with inflammation [33, 34], and by gain-of-function mutations in the intracellular viral RNA sensors, MDA5 and RIG-I, respectively [35]. Therefore, precise regulation of IFNβ production is crucial to prevent the unnecessary spread of inflammation. Optineurin may play the physiologically important role of suppressing IFNβ overproduction through negative feedback.

This study uncovered a mechanism of IFNβ overproduction in virus-infected *Optn*-KO cells. Specifically, optineurin deficiency causes a reduction of autophagic activity, which in turn leads to inadequate clearance of viral genome components from an infection. The accumulated viral RNA subsequently induces excessive IFNβ production. Therefore, we suggest that optineurin has an important role in the elimination of viral components containing viral RNA from host cells. Although it remains unclear how optineurin captures viral components, it might bind viral nucleic acids directly through its leucine zipper and zinc finger domains or indirectly through viral proteins composed of nucleocapsid. The *interferon beta* gene is conserved in species that have evolved from vertebrates; however, optineurin is conserved in more evolutionarily ancient species, such as nematodes and plants [36, 37]. Therefore, autophagic machinery involving optineurin is considered to be a relatively primitive mechanism for eliminating foreign substances from cells. Autophagy receptors must therefore be evolutionarily conserved, and they must have acquired redundancy because they are necessary for cellular homeostasis and pathogen elimination. Impaired autophagy receptors may make it more difficult to keep cells healthy because of the accumulation of unwanted components from foreign pathogens and host metabolism. This study revealed that viral infection of cells deficient in optineurin, sequestosome 1 or NDP52 produce IFNβ excessively. These three molecules have similar domains and function as autophagy receptors [13]; therefore, autophagy receptors are expected to be important for viral elimination. Also, as mentioned above, *Optn*-KO cells and mice were apparently indistinguishable from WT controls under normal conditions. Therefore, these findings indicate that the function of optineurin is compensated for by other autophagy receptors. This redundancy is thought to explain the minor difference in survival rates between *Optn*-KO and WT mice in our viral challenges.

This study demonstrates that IFNβ overproduction in response to viral infection is observed in ALS patient cells carrying mutations in *OPTN* but not in other ALS-causative genes. Therefore, this phenomenon of IFNβ overproduction in response to viral infection is not common to all forms of ALS. However, this phenomenon was observed in cells with a mutation in *SQSTM1*, an ALS-causative gene that encodes sequestosome 1, an autophagy receptor. More precisely, *Sqstm1*(+/−) cells overproduce IFNβ during a viral infection, but this

is not the case in *Sqstm1*(−/−) cells. The reason why *Sqstm1*(−/−) cells do not overproduce IFNβ remains unclear. However, *Sqstm1*(−/−) cells readily induced cell death compared with *Sqstm1*(+/−) cells during viral infection. Therefore, we speculate that *Sqstm1*(−/−) cells produce less IFNβ because cell death occurs more readily. Interestingly, all cases of *SQSTM1*-related ALS are heterozygous for mutations in *SQSTM1*; homozygosity for mutations in *SQSTM1*(−/−) causes a childhood-onset neurodegenerative disorder [38, 39]. In addition, optineurin and sequestosome 1 bind to TBK1, which is a critical molecule in the IFNβ production pathway [13] and an ALS-causative gene [40, 41]. Collectively, these results indicate that excessive levels of IFNβ and autophagy receptors may contribute to ALS.

ALS generally occurs in middle age or later, and patients live normal lives before disease onset [42]. In addition, despite many ALS-causative genes having been found by the analysis of patient genomes, few genetically modified mice reproduce ALS symptoms [43]. This indicates that other factors are required for ALS onset. This study shows that in the non-infected state, *Optn*-KO MEFs do not differ from WT cells. Furthermore, *Optn*-KO mice show no difference in physical appearance, behaviour, or lifespan compared with WT mice [21]. However, in this study, viral infection revealed a difference between optineurin-deficient and WT groups with respect to levels of IFNβ production. Enterovirus has been detected in the central nervous system of ALS patients [44]; therefore, viral infection of neurons in optineurin-deficient individuals might be a trigger for ALS development.

## Supporting information

**S1 Fig. Optineurin is involved in IFNβ production following viral infection *in vitro* and the characteristics of *Optn*-KO MEFs.** (A–C) Transcriptional activities of NF-κB ($n$ = 3), IRF3 ($n$ = 3), and the *IFNB* promoter ($n$ = 3) in cells transfected with optineurin-expression or empty vectors 12 hours after viral infection of SeV (Cantell strain) were measured by luciferase assays. (D) *Optn*-KO and WT MEFs in steady state culture. Scale bars, 50 μm. (E and F) Cell growth and viability of *Optn*-KO and WT MEFs in steady state culture were measured by counting live and dead cells stained with trypan blue at the indicated days. $n$ = 3 independent replicates per time point. (G) Relative viral DI genome copy numbers in WT ($n$ = 4) and *Optn*-KO ($n$ = 4) MEFs infected with SeV (Cantell strain) were measured by qPCR at 6 hours after inoculation. Data are presented as mean values ± SD. Two-tailed unpaired Welch's $t$-test (A), two-tailed unpaired Student's $t$-test (B, C, E, F, and G). *p < 0.05, **p < 0.01, ***p < 0.001. (TIF)

**S2 Fig. Optineurin deficiency is involved in autophagy failure.** (A) (upper panels) Fluorescence images of WT and *Optn*-KO MEFs stained with the autophagy marker, CYTO-ID, under starvation conditions. Scale bars, 50 μm. (lower panel) Fluorescence intensities of WT and *Optn*-KO MEFs stained with CYTO-ID under starvation conditions were measured by flow cytometry and the results were compared. The black line indicates the unstained control. (B) LC3-I, LC3-II, optineurin, and actin of WT and *Optn*-KO MEFs under starvation conditions were examined by western blotting. (C) Optineurin protein levels in parental and two *Optn*-KO clones of GFP-LC3-RFP-LC3ΔG MEFs were examined by western blotting. (TIF)

**S3 Fig. Assessment of mouse primary astrocytes.** Fluorescence images of primary cells isolated from WT and *Optn*-KO mouse pups. The cells were stained with antibodies against the indicated proteins to confirm purity. Scale bars, 50 μm. (TIF)

**S4 Fig. IFNβ production by cells from several ALS patients following viral infection.** (A) Transcriptional activity of the *IFNB* promoter in healthy donor ($n = 4$) and ALS-optineurin: pQ398* patient ($n = 4$) fibroblasts was measured by luciferase assays at 24 hours after viral inoculation. (B) IFNβ production by healthy donor and ALS-optineurin:p.Q398* patient fibroblasts at the indicated hours after inoculation. $n = 3$ independent replicates per group at the indicated times. (C) IFNβ production from ALS patient fibroblasts carrying a mutation in the indicated causative genes ($n = 3$). The patient fibroblasts were infected with SeV (Cantell strain) or mock treated. Twenty-four hours after inoculation, IFNβ in culture medium was measured by ELISA. Data are presented as mean values ± SD. Two-way ANOVA followed by the Tukey–Kramer method (A and B) and one-way ANOVA followed by Dunnett's test (C) were applied for statistical analyses. *$p < 0.05$, **$p < 0.01$, ***$p < 0.001$.
(TIF)

**S5 Fig. Viral infection to *Optn*-KO medaka.** (A) Optineurin protein levels in cells isolated from WT and *Optn*-KO medaka (*Oryzias latipes*) were examined by western blotting. (B) Survival rates of WT ($n = 60$) and *Optn*-KO ($n = 60$) medaka infected with betanodavirus. The Kaplan–Meier method was applied for statistical analyses. ***$p < 0.001$.
(TIF)

**S6 Fig. High levels of cell death of *Optn*-KO cells following viral infection.** WT and *Optn*-KO MEFs infected with SeV (Z strain) 48 hours after inoculation and influenza virus (PR8 strain) 24 hours after inoculation. Scale bars, 100 mm.
(TIF)

**S7 Fig. Profiles of cytokine expression by ALS patient cells and mouse neural cells after viral infection.** (A) Expression of 36 human cytokines by healthy donor and ALS-optineurin: p.Q398* patient fibroblasts infected with SeV (Cantell stain) or mock treated were examined by Proteome Profiler Antibody Arrays. (B) Expression of 40 mouse cytokines from WT and *Optn*-KO primary astrocytes infected with SeV (Cantell stain) or mock treated were examined by Proteome Profiler Antibody Arrays.
(TIF)

## Acknowledgments

We thank Drs. Noboru Mizushima and Toru Yanagawa for providing GFP-LC3-RFP-LC3ΔG and *Sqstm1* (+/+), (+/−), and (−/−) MEFs, respectively. We also thank Masaaki Komatsu, Tetsuya Saito, Teruhiko Hatakeyama, Tsuyoshi Tanaka, Masaya Matsumoto, Yoko Hayashi, Ryoko Kawabata, and Reiko Yoshimoto for their help with experiments, as well as Yoshihito Taniguchi, Izumi Hide, Ichiro Takahashi, Norimitsu Morioka, and Masahiro Fujii for their advice. We are grateful to the National Institute of Infectious Diseases in Japan, NBRP medaka in Japan (https://shigen.nig.ac.jp/medaka/), the RIKEN BRC in the National BioResource Project of the MEXT/AMED in Japan, and the Coriell Institute in the US for providing influenza virus, *Optn*-KO medaka, GFP-LC3-RFP-LC3ΔG MEFs, and ALS patient cells, respectively. We also thank Mitchell Arico and other editors from Edanz (https://jp.edanz.com/ac) for editing drafts of this manuscript.

## Author Contributions

**Conceptualization:** Masaya Fukushi, Takemasa Sakaguchi.

**Data curation:** Masaya Fukushi.

**Formal analysis:** Masaya Fukushi.

**Funding acquisition:** Masaya Fukushi, Ryosuke Ohsawa, Yasushi Okinaka, Takashi Irie, Hideshi Kawakami, Takemasa Sakaguchi.

**Investigation:** Masaya Fukushi, Ryosuke Ohsawa, Yasushi Okinaka, Tohru Kiyono, Masaya Moriwaki, Takashi Irie, Kosuke Oda, Yasuhiro Kamei.

**Resources:** Daisuke Oikawa, Fuminori Tokunaga, Yusuke Sotomaru, Hirofumi Maruyama, Hideshi Kawakami, Takemasa Sakaguchi.

**Supervision:** Hideshi Kawakami, Takemasa Sakaguchi.

**Validation:** Hideshi Kawakami, Takemasa Sakaguchi.

**Writing – original draft:** Masaya Fukushi.

**Writing – review & editing:** Masaya Fukushi, Ryosuke Ohsawa, Hideshi Kawakami, Takemasa Sakaguchi.

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
