## [Decision Letter · Decision Letter 0]

14 Apr 2023

PONE-D-23-07077Optineurin deficiency impairs autophagy to cause interferon beta overproduction and increased survival of mice following viral infectionPLOS ONE

Dear Dr. Fukushi,

Thank you for submitting your manuscript to PLOS ONE. After careful consideration, we feel that it has merit but does not fully meet PLOS ONE’s publication criteria as it currently stands. Therefore, we invite you to submit a revised version of the manuscript that addresses the points raised during the review process.

We look forward to receiving your revised manuscript.

Kind regards,

Donna A. MacDuff, Ph.D

Academic Editor

PLOS ONE

Journal Requirements:

    "We thank Drs. Noboru Mizushima and Toru Yanagawa for providing GFP-LC3-RFP594 LC3�G and Sqstm1 (+/+), (+/−), and (−/−) MEFs, respectively. We also thank Masaaki Komatsu, Tetsuya Saito, Teruhiko Hatakeyama, Tsuyoshi Tanaka, Masaya Matsumoto, Yoko Hayashi, Ryoko Kawabata, and Reiko Yoshimoto for their help with experiments, as well as Yoshihito Taniguchi, Izumi Hide, Ichiro Takahashi, Norimitsu Morioka, and Masahiro Fujii for their advice. We are grateful to the National Institute of Infectious Diseases in Japan, NBRP medaka in Japan (https://shigen.nig.ac.jp/medaka/), the RIKEN BRC in the National BioResource Project of the MEXT/AMED in Japan, and the Coriell Institute in the US for providing influenza virus, Optn-KO medaka, GFP-LC3-RFP602 LC3�G MEFs, and ALS patient cells, respectively. We also thank Mitchell Arico and other editors from Edanz (https://jp.edanz.com/ac) for editing drafts of this manuscript. 

Some experiments in this study were carried out at the Natural Science Center for Basic Research and Development, Hiroshima University. This study was supported by JSPS KAKENHI Grant Numbers JP16K08812, JP25460568, JP26242085, JP19K22968,JP26830035 and JP21K07461, and by the Tsuchiya Memorial Medical Foundation, the Program for Promotion of Basic and Applied Research for Innovations in Biooriented Industry (BRAIN), and the Takeda Science Foundation. This study was partially supported by the NIBB Individual Collaborative Research Program (ID: 12-361 and 12-340) and the Program of the Network-type Joint Usage/Research Center for Radiation Disaster Medical Science"

Please note that funding information should not appear in any section or other areas of your manuscript. We will only publish funding information present in the Funding Statement section of the online submission form. Please remove any funding-related text from the manuscript.

Additional Editor Comments:

Since several studies have reported findings contradictory to those presented in this manuscript, please include a discussion of those studies and possible reasons for the discrepancies. Please also provide information about the impact of the human Q398* mutation on protein function. 

Reviewers' comments:

Reviewer's Responses to Questions

**Comments to the Author**

1. Is the manuscript technically sound, and do the data support the conclusions?

Reviewer #1: Partly

Reviewer #2: Yes

2. Has the statistical analysis been performed appropriately and rigorously? 

Reviewer #1: I Don't Know

Reviewer #2: Yes

3. Have the authors made all data underlying the findings in their manuscript fully available?

Reviewer #1: Yes

Reviewer #2: Yes

4. Is the manuscript presented in an intelligible fashion and written in standard English?

Reviewer #1: Yes

Reviewer #2: Yes

5. Review Comments to the Author

Reviewer #1: This is an interesting article that tries to show that OPTN helps degrade viral defective genomes inturn not activating IFN-b production during Sendai and Flu virus infections. On the flip side, excessive genome presence in the OPTN KO creates more IFNb, leading to an antiviral response.

The manuscript makes good sense based on the hypothesis stated but is not supported by the data.

CRITICAL Expt that need to be done:

1. Authors need to show infectious viral titer during in vitro experiments

2. Authors need to show defective viral genome in vivo experiments.

If there is more virus in vitro or less defective genomes in vivo, their hypothesis falls apart.

Additional details are mentioned in the PDF document attached.

Reviewer #2: In this manuscript, the authors report that optineurin deficiency promotes an increased IFNb production following viral infection both in vitro and in vivo. This IFNb overproduction in optineurin deficient cells is proposed to be caused by an accumulation of viral nucleic acid as a consequence of a reduced autophagy since optineurin is involved in autophagy.

There is a discrepancy about the role of optineurin in innate immunity. Its role is highly debated in the context of innate immunity. While here, the results suggest that optineurin dampens IFNb production after viral infection through the autophagic degradation of viral nucleic acids, at least 5 previous studies (PMID: 24244017, 27086836, 27538435, 26677802, 21862579) have reported that optineurin is in contrast required for the signaling leading to the production of IFNb both in vitro and in vivo. How do the authors explain this?

Based on their results, at the molecular level, do the authors observe an increased phosphorylation of TBK1 and IFR3 in optineurin deficient cells following viral infection? Moreover, the authors have also analyzed cells from patients with mutations in optineurin which is really different to cells fully deficient in optineurin since mutations may impair some of its function but not all the functions like a full knock down.

In fig 4C, why only Sqstm1+/- cells overproduces IFNb but not -/- cells?

6. PLOS authors have the option to publish the peer review history of their article (what does this mean?). If published, this will include your full peer review and any attached files.

Reviewer #1: **Yes: **Tejabhiram Yadavalli

Reviewer #2: **Yes: **Damien Arnoult

---

## [Author Response · Author response to Decision Letter 0]

31 May 2023

I. Response to Additional Editor Comments

Thank you for your valuable comments and suggestions. Our manuscript has been improved in response to the comments we received. Our point-by-point responses to each of the comments are listed below.

1. Since several studies have reported findings contradictory to those presented in this manuscript, please include a discussion of those studies and possible reasons for the discrepancies.

Response: Thank you for your comment. We have rewritten the Discussion section of the revised manuscript to describe the discrepancies between studies and provide possible explanations, as follows:

‘Regarding IFN� production in optineurin-deficient cells, there are discrepancies between different reports (PMID: 24244017, 27086836, 26677802, 21862579, 20174559 and 25923723). We attribute these discrepancies to the cell types, stimuli, and methods used in the studies. In brief, some groups used bone marrow-derived macrophages (BMDM), LPS, and poly(I:C). LPS and poly(I:C) were directly added to the culture media of BMDM, which express Toll-like receptors (TLRs). Therefore, the stimuli of LPS and poly(I:C) are thought to be transmitted into cells through TLR 4 located on the cell surface or TLR 3 located in endosome. In our study, fibroblasts were mainly used that do not express TLRs, and viral infection was used as a stimulus. Viruses directly enter into the cytoplasm, and their genomes are recognised by cytoplasmic viral RNA sensors, such as retinoic acid-inducible gene I (RIG-I) and melanoma differentiation-associated protein 5 (MDA5), not by TLRs. Other groups also reported excess IFN� production in optineurin-deficient fibroblasts during viral infection (PMID: 20174559 and 25923723). On the basis of these findings, we propose that the discrepancy in IFN� production in optineurin-deficient cells is caused by a difference in the signalling pathways induced in response to stimuli.’

In addition, an in vivo study by Slowicka et al. (Eur. J. Immunol., 2016, PMID: 26677802) showed that mortality was slightly higher in OPTN knockout mice (4 of 7 mice died) compared with control mice (2 of 7 mice died). However, in their report, only 7 mice were used, and no statistically significant difference was found. By contrast, sufficient numbers of mice were used in our in vivo infection experiments of both SeV infection (Optn-KO, n = 40; WT, n = 39) and influenza virus infection (Optn-KO, n = 44; WT, n = 50), and significant differences were shown by statistical analysis. Therefore, we propose that the in vivo differences between our results and that of the other group may be a result of the number of animals used in the experiments.

2. Please also provide information about the impact of the human Q398* mutation on protein function. 

Response: A previous study by our group showed that optineurin protein itself is not produced in ALS-optineurin:p.Q398* cells by a mechanism thought to be nonsense-mediated mRNA decay (Maruyama et al., Nature 2010, Supplementary Figure 3a, PMID: 20428114). Therefore, ALS-optineurin:p.Q398* cells are thought to show a similar response to that of Optn-KO cells.

II. Response to the Reviewer #1’s comments

1. Authors need to show infectious viral titer during in vitro experiments

Response: We apologize for any misunderstanding relating to our inadequate description of the SeV DI genome. The DI genome is specific to the Cantell strain of SeV and is not present in the SeV Z strain or influenza virus (Yoshida et al. J. Virology 2018, PMID: 29237838). It is also worth noting that the DI genome is replicated more rapidly than the full genome of SeV because it is shorter in length. IFN� is known to be induced in response to viral infection. SeV Cantell strain is broadly used in studies of IFN� production during viral infection because the DI genome of this strain is a strong inducer of IFN�. However, IFN� is also induced in response to infection by SeV Z strain that do not have DI genome in both in vitro and in vivo (Figs 1G and 5B). As pointed out by the reviewer’s comment, we were also interested in the viral titre of SeV. Therefore, we assessed the quantities of full viral genome in infected cells and the quantity was found to be higher in infected Optn-KO cells than in WT cells, similar to findings with the viral DI genome (Fig 1H).

We did not measure infectious viral titre of SeV in our in vitro experiments. This was because the measurement of infectious viral titre at 24 hours after inoculation was below the limit of detection. Instead, in this study, RT-qPCR was used to measure viral RNA levels in order to know viral replication (Fig 1H). In addition, Optn-KO cells showed more pronounced cell death compared with WT cells after 24 hours post-inoculation (S6 Fig). Therefore, we think that it is difficult to draw comparisons between the quantities of infectious virus particles produced by Optn-KO and WT cells after 24 hours post-inoculation. As shown in Fig 1H, both viral DI and full genomes were highly accumulated in Optn-KO cells. The reason of large quantities of both genomes in Optn-KO cells was shown to be caused by its low autophagic activity (Fig 2). However, we think that there is not necessarily a parallel between the large quantity of viral genomes inside the cells and the large quantity of infectious viral particles outside the cells. Because, from our in vivo study, the viral titre in Optn-KO mice was low, compared with WT mice. We speculate that the reason is that large amount of IFN� strongly inhibits viral spread and also strongly leads to cell death of the infected cells.

2. Authors need to show defective viral genome in vivo experiments.

Response: In our mouse experiments, Z strain of SeV and influenza virus (PR8 strain) were used. Z strain and influenza virus do not possess a DI genome, as mentioned above. Therefore, we did not measure the quantity of DI genome in our mouse experiments. The reason why SeV Cantell strain was not used in in vivo experiments is that inoculation with the maximum titre of Cantell strain of SeV resulted in no change in the general condition of the animals, as mentioned in our manuscript. We showed that SeV Z strain also induced IFN� overproduction in vitro and in vivo (Figs 1G and 5B). Therefore, IFN� production is induced not only in response to DI genome but also full genome.

3. This report makes a generalized claim that contradicts other published data regarding OPTN and viruses. The authors can only claim that this data is true for Sendai Virus (This reviewer 1’s comment is attached on top page of PDF file of the original manuscript).

Response: Thank you for your comment. We have rewritten the Discussion section of the revised manuscript to describe the discrepancies between studies and provide possible explanations, as follows:

‘Regarding IFN� production in optineurin-deficient cells, there are discrepancies between different reports (PMID: 24244017, 27086836, 26677802, 21862579, 20174559 and 25923723). We attribute these discrepancies to the cell types, stimuli, and methods used in the studies. In brief, some groups used bone marrow-derived macrophages (BMDM), LPS, and poly(I:C). LPS and poly(I:C) were directly added to the culture media of BMDM, which express Toll-like receptors (TLRs). Therefore, the stimuli of LPS and poly(I:C) are thought to be transmitted into cells through TLR 4 located on the cell surface or TLR 3 located in endosome. In our study, fibroblasts were mainly used that do not express TLRs, and viral infection was used as a stimulus. Viruses directly enter into the cytoplasm, and their genomes are recognised by cytoplasmic viral RNA sensors, such as retinoic acid-inducible gene I (RIG-I) and melanoma differentiation-associated protein 5 (MDA5), not by TLRs. Other groups also reported excess IFN� production in optineurin-deficient fibroblasts during viral infection (PMID: 20174559 and 25923723). On the basis of these findings, we propose that the discrepancy in IFN� production in optineurin-deficient cells is caused by a difference in the signalling pathways induced in response to stimuli.’

In addition, we used several viruses besides SeV Cantell and Z strains, including influenza virus and fish viruses. Therefore, the phenomena reported in this study do not appear to be limited to SeV infection.

4. Please add Sendai and Influenza virus somewhere in the abstract (This reviewer 1’s comment is attached at line 56 on page 4 of the original manuscript).

Response: We used several viruses besides SeV Cantell and Z strains, including influenza virus and fish viruses, as well as, several types of cells, and mice and fish. We have rewritten the Methods section of the Abstract in the revised manuscript to convey this.

5. Show IFN B levels during poly I:C transfection (This reviewer 1’s comment is attached at line 331 on page 19 of the original manuscript).

Response: Poly(I:C) is historically and broadly used as an IFN� inducer instead of viral infection in virology and immunology fields. In this study, virus was infected to several types of optineurin-disrupted cells in order to measure IFN� production. Because poly(I:C) transfection is more artificial than viral infection, we did not carry out a poly(I:C) transfection experiment for the measurement of IFN� production. However, we speculate that IFN� production in poly(I:C)-transfected Optn-KO MEF is greater than that of WT cells. In addition, Mankouri et al. (PLoS Path., 2010, PMID: 20174559) reported that optineurin-disrupted cells using siRNA produced excess IFN� by poly(I:C) compared with WT cells. Therefore, to also avoid duplication, we did not perform poly(I:C) transfection to Optn-KO MEFs regarding IFN� production.

6. Please show viral titers in the cells through plaque assay. Defective viral genome is shown but not infectious titer (This reviewer 1’s comment is attached at line 345 on page 20 of the original manuscript).

Response: We examined the quantity of SeV full genome by RT-qPCR to determine the viral titre in the infected cells, and the full genome quantity was higher in Optn-KO MEFs compared with WT cells, similar to the findings with DI genome (Fig 1H). Both the DI and full genomes of SeV are known to induce IFN� production and, therefore, large amounts of DI and full genomes result in the excessive induction of IFN� in Optn-KO MEFs.

7. Please show the DI genome quantities in mouse lungs (This reviewer 1’s comment is attached at line 474 on page 27 of the original manuscript).

Response: In our mouse experiments, Z strain of SeV and influenza virus (PR8 strain) were used. Z strain and influenza virus do not possess a DI genome, as mentioned above. Therefore, we did not measure the quantity of DI genome in our mouse experiments. The reason why SeV Cantell strain was not used in in vivo experiments is that inoculation with the maximum titre of Cantell strain of SeV resulted in no change in the general condition of animals, as mentioned in our manuscript. We showed that SeV Z strain also induced IFN� overproduction in vitro and in vivo (Figs 1G and 5B). Therefore, IFN� production is induced not only in response to DI genome but also full genome.

8. Please explain if the defective viral genome is lower in OPTN KO cells. It is important to show that OPTN KO cells have higher defective viral genome but lower infectious viral titer (This reviewer 1’s comment is attached at line 476 on page 27 of the original manuscript).

Response: In this study, the quantities of both DI and full genomes in Optn-KO cells were greater than those of WT cells (Fig 1H). These large quantities of both viral genomes induced excessive levels of IFN� production, and IFN� is known to cause apoptosis in infected cells (Tanaka et al., Genes Cells, 1998, PMID: 9581980; Taniguchi et al., Curr Opin Immunol., 2002, PMID: 11790540). In fact, we observed higher levels of cell death in Optn-KO cells than WT cells following viral infection (S6 Fig). Taken together with our in vitro and in vivo results, we conclude that the quantity of viral genomes in Optn-KO cells was greater than in WT cells (Fig 1H), and viral titres in the lungs of Optn-KO mice were lower than those of WT mice (Fig 5 E and F). In addition, IFN� overproduction and pronounced cell death were observed in Optn-KO cells and mice. Large quantities of viral genomes in optineurin-deficient cells during viral infection induced IFN� overproduction, and excess IFN� significantly enhanced the death of infected cells and the anti-viral state of uninfected cell. As a consequence, this strongly suppressed the spread of the virus. Strong inhibition of viral spread led to the improved survival of infected Optn-KO animals.

III. Response to the Reviewer #2’s comments

1. There is a discrepancy about the role of optineurin in innate immunity. Its role is highly debated in the context of innate immunity. While here, the results suggest that optineurin dampens IFNb production after viral infection through the autophagic degradation of viral nucleic acids, at least 5 previous studies (PMID: 24244017, 27086836, 27538435, 26677802, 21862579) have reported that optineurin is in contrast required for the signaling leading to the production of IFNb both in vitro and in vivo. How do the authors explain this? Based on their results, at the molecular level, do the authors observe an increased phosphorylation of TBK1 and IFR3 in optineurin deficient cells following viral infection?

Response: Thank you for your comment. We have rewritten the Discussion section of the revised manuscript to describe the discrepancies between studies and provide possible explanations, as follows:

‘Regarding IFN� production in optineurin-deficient cells, there are discrepancies between different reports (PMID: 24244017, 27086836, 26677802, 21862579, 20174559 and 25923723). We attribute these discrepancies to the cell types, stimuli, and methods used in the studies. In brief, some groups used bone marrow-derived macrophages (BMDM), LPS, and poly(I:C). LPS and poly(I:C) were directly added to the culture media of BMDM, which express Toll-like receptors (TLRs). Therefore, the stimuli of LPS and poly(I:C) are thought to be transmitted into cells through TLR 4 located on the cell surface or TLR 3 located in endosome. In our study, fibroblasts were mainly used that do not express TLRs, and viral infection was used as a stimulus. Viruses directly enter into the cytoplasm, and their genomes are recognised by cytoplasmic viral RNA sensors, such as retinoic acid-inducible gene I (RIG-I) and melanoma differentiation-associated protein 5 (MDA5), not by TLRs. Other groups also reported excess IFN� production in optineurin-deficient fibroblasts during viral infection (PMID: 20174559 and 25923723). On the basis of these findings, we propose that the discrepancy in IFN� production in optineurin-deficient cells is caused by a difference in the signalling pathways induced in response to stimuli.’

Although a part of the results showed by Pourcelot et al. (PMID: 27538435) that you indicated is also contrary to our results, we cannot find suitable reasons to this discrepancy except virus strains. However, Mankouri et al. (PLoS Path., 2010, PMID: 20174559) and Génin et al. (PLoS Path., 2015, PMID: 25923723) showed excess IFN� production in optineurin-deficient fibroblasts during viral infection similar to our results.

In addition, an in vivo study by Slowicka et al. (Eur. J. Immunol., 2016, PMID: 26677802) showed that mortality was slightly higher in OPTN knockout mice (4 of 7 mice died) compared to control mice (2 of 7 mice died). However, in their report, only 7 mice were used, and no statistically significant difference was found. By contrast, sufficient numbers of mice were used in our in vivo infection experiments of both SeV infection (Optn-KO, n = 40; WT, n = 39) and influenza virus infection (Optn-KO, n = 44; WT, n = 50), and significant differences were shown by statistical analysis. Therefore, we propose that the in vivo differences between our results and that of the other group may be a result of the number of animals used in the experiments.

 In our study, we did not determine whether TBK1 and IRF3 are phosphorylated in optineurin-deficient cells following viral infection. However, IRF3 activity in our luciferase assay was elevated in Optn-KO cells during viral infection compared with uninfected cells (Fig 1C). Therefore, we think that IRF3 is phosphorylated in Optn-KO cells during viral infection. In addition, IRF3 activity was inhibited in optineurin-overexpressed cells during SeV infection (S1 Fig B). From these results, we think that IRF3 is functional in Optn-KO cells, and that there is a relationship between optineurin and IRF3 activity.

2. Moreover, the authors have also analyzed cells from patients with mutations in optineurin which is really different to cells fully deficient in optineurin since mutations may impair some of its function but not all the functions like a full knock down.

Response: A previous study by our group showed that optineurin protein itself is not produced in ALS-optineurin:p.Q398* cells by a mechanism thought to be nonsense-mediated mRNA decay (Maruyama et al., Nature 2010, Supplementary Figure 3a, PMID: 20428114). Therefore, ALS-optineurin:p.Q398* cells are thought to show a similar response to that of Optn-KO cells.

3. In fig 4C, why only Sqstm1+/- cells overproduces IFNb but not -/- cells? 

Response: We are also interested in this finding, but currently have no direct evidence to explain this. However, we have added a possible explanation to the Discussion section, as follows:

‘The reason why Sqstm1(˗/˗) cells do not overproduce IFN� remains unclear. However, Sqstm1(˗/˗) cells readily induced cell death compared with Sqstm1(+/˗) cells during viral infection. Therefore, we speculate that Sqstm1(˗/˗) cells produce less IFN� because cell death occurs more readily.’

---

## [Decision Letter · Decision Letter 1]

7 Jun 2023

Optineurin deficiency impairs autophagy to cause interferon beta overproduction and increased survival of mice following viral infection

PONE-D-23-07077R1

Dear Dr. Fukushi,

We’re pleased to inform you that your manuscript has been judged scientifically suitable for publication and will be formally accepted for publication once it meets all outstanding technical requirements.

Kind regards,

Donna A. MacDuff, Ph.D

Academic Editor

PLOS ONE

Additional Editor Comments (optional):

Reviewers' comments:

Reviewer's Responses to Questions

**Comments to the Author**

1. If the authors have adequately addressed your comments raised in a previous round of review and you feel that this manuscript is now acceptable for publication, you may indicate that here to bypass the “Comments to the Author” section, enter your conflict of interest statement in the “Confidential to Editor” section, and submit your "Accept" recommendation.

Reviewer #1: All comments have been addressed

Reviewer #2: (No Response)

2. Is the manuscript technically sound, and do the data support the conclusions?

Reviewer #1: Partly

Reviewer #2: Yes

3. Has the statistical analysis been performed appropriately and rigorously? 

Reviewer #1: Yes

Reviewer #2: Yes

4. Have the authors made all data underlying the findings in their manuscript fully available?

Reviewer #1: No

Reviewer #2: Yes

5. Is the manuscript presented in an intelligible fashion and written in standard English?

Reviewer #1: Yes

Reviewer #2: Yes

6. Review Comments to the Author

Reviewer #1: Acceptable in current format. It would be great if the raw data of viral genome counts and infectious viral titer (plaque assay?) could be provided with the final submission

Reviewer #2: The referees' comments have been addressed in a satisfactory manner, I therefore recommend acceptance for publication.

7. PLOS authors have the option to publish the peer review history of their article (what does this mean?). If published, this will include your full peer review and any attached files.

Reviewer #1: No

Reviewer #2: **Yes: **Damien Arnoult

---

## [Editor Report · Acceptance letter]

13 Jun 2023

PONE-D-23-07077R1 

Optineurin deficiency impairs autophagy to cause interferon beta overproduction and increased survival of mice following viral infection 

Dear Dr. Fukushi:

I'm pleased to inform you that your manuscript has been deemed suitable for publication in PLOS ONE. Congratulations! Your manuscript is now with our production department. 

Kind regards, 

on behalf of

Dr. Donna A. MacDuff 

Academic Editor

PLOS ONE